# The Role of Prebiotic and Herbal Supplementation in Enhancing Welfare and Resilience of Kenguri Sheep Subjected to Transportation Stress

**DOI:** 10.3390/vetsci12050442

**Published:** 2025-05-05

**Authors:** Veerasamy Sejian, Chinnasamy Devaraj, Chikamagalore Gopalakrishna Shashank, Mullakkalparambil Velayudhan Silpa, Artabandhu Sahoo, Raghavendra Bhatta

**Affiliations:** 1Centre for Climate Resilient Animal Adaptation Studies, ICAR-National Institute of Animal Nutrition and Physiology, Adugodi, Bangalore 560030, India; drcdeva@gmail.com (C.D.); shanko009@gmail.com (C.G.S.); sahooarta1@gmail.com (A.S.); ragha0209@yahoo.com (R.B.); 2Centre for Translational Research, Rajiv Gandhi Institute of Veterinary Education and Research, Kurumbapet 605009, India; mv.silpa@gmail.com

**Keywords:** body temperature, heat shock proteins, hematocrit, hemoglobin, respiration rate

## Abstract

**Simple Summary:**

A study was conducted to assess the efficacy of prebiotic and herbal supplements to relieve transportation stress in Kenguri sheep. The weather variables recorded in the study indicated that the Kenguri ewes were subjected to heat stress during transportation. The low levels of stress markers such as respiration rate, pulse rate, hemoglobin, and hematocrit in both the prebiotic and herbal supplemented groups as compared to the control (non-supplemented) group clearly demonstrate the transportation-stress-mitigating potential of both prebiotic and herbal supplements. These results provide a scientific basis for incorporating such supplements into livestock management practices, especially in arid and semi-arid regions, where transportation stress is exacerbated by high temperatures.

**Abstract:**

A study was conducted to assess the efficacy of prebiotic and herbal supplements to relieve transportation stress based on changes in physiological, hematological, and molecular responses in Kenguri sheep. Thirty healthy female sheep were randomly divided into three groups: a control group (CKS) with no supplementation, a prebiotic supplementation group (PKS), and an herbal supplementation group (HKS). The animals were transported 230 km over seven hours during summer conditions, with temperatures ranging from 32.5 °C to 34.9 °C. The groups that received the prebiotic (75.6 breaths/min; 64.8 beats/min) and herbal supplementation (31.0 breaths/min; 66.8 beats/min) had a significantly reduced respiration rate (RR) and pulse rate (PR) compared to those of the control group (38.7 breaths/min; 75.6 beats/min) (*p* < 0.01 and *p* < 0.05, respectively), indicating improved physiological stability. The hemoglobin (HGB) and hematocrit (HCT) levels were also significantly lower in the PKS (24.2 g/dL; 24.8%) and HKS (24.7 g/dL; 24.5%) groups than in the CKS (28.1 g/dL; 24.9%) (*p* < 0.05), highlighting the mitigation of hematological stress. Further, the plasma glucose level was significantly higher (*p* < 0.01) in the HKS group (80.0 mg/dL) compared to the CKS group (63.5 mg/dL). However, rectal temperature (RT) and skin temperature (ST), red blood cells (RBCs), mean corpuscular volume (MCV), and white blood cells (WBCs) showed no significant differences among the groups. These findings demonstrate that prebiotic and herbal supplementation can effectively reduce transportation-induced stress in Kenguri sheep, offering a practical strategy to improve the welfare and resilience of livestock under challenging environmental conditions.

## 1. Introduction

Livestock transportation is a crucial component of modern animal husbandry, facilitating trade, breeding, and slaughter activities. However, it often imposes significant stress on animals, adversely affecting their welfare and productivity [1]. Transportation stress arises from multiple factors, including physical confinement, vibration, noise, overcrowding, and exposure to extreme environmental conditions. These stressors trigger a cascade of physiological and behavioral responses that compromise the health and performance of animals [2].

Ruminants, such as sheep and goats, are particularly vulnerable to transportation stress due to their unique digestive systems and social behaviors [3]. Their digestive system, particularly the rumen, is sensitive to disruptions, and their social behavior makes them vulnerable to separation stress. Further, transportation can disrupt the rumen’s microbial balance, affecting digestion and potentially leading to health issues in small ruminants. Small ruminants are social animals and form strong social bonds within their groups. Thus, separation from their social group during transport can cause significant stress, especially in sheep and goats.

Physiological responses include elevated cortisol levels, increased heart rates, and immune suppression, all of which disrupt homeostasis and can lead to reduced feed intake and impaired rumen function [1,4]. Behaviorally, animals may exhibit restlessness, vocalizations, and decreased social cohesion, indicating heightened stress levels. These changes not only affect animal welfare but also result in economic losses due to decreased weight gain, poor meat quality, and higher mortality rates [3].

Kenguri sheep, an indigenous breed in India, are well adapted to arid climates and are integral to the livelihoods of small-scale farmers. Known for their resilience, they are a cornerstone of sustainable livestock farming systems in the region [5]. Despite their hardiness, Kenguri sheep are not immune to transportation stress, particularly during the summer months when high temperatures exacerbate the adverse effects of transport. Heat stress during transportation can lead to hyperthermia, dehydration, and oxidative damage, further compromising the animals’ ability to cope with stressors [3]. Addressing transportation stress in Kenguri sheep is crucial for improving animal welfare and ensuring economic sustainability for farmers.

Several measures have been employed to mitigate transportation stress, ranging from physical adjustments to pharmaceutical interventions. Common physical measures include optimizing vehicle design to ensure adequate ventilation, reducing stocking density, and providing access to water and feed during transit. Pharmaceutical approaches, such as the administration of sedatives and anti-inflammatory drugs, have also been explored [6,7]. While these strategies offer some relief, they often have significant limitations. Physical measures may not fully address the underlying physiological disruptions, and pharmaceutical interventions are costly and potentially impractical for small-scale farmers. Additionally, concerns about drug residues in meat products and the side effects of pharmaceuticals highlight the need for alternative approaches [8].

Recently, attention has turned toward herbal and prebiotic supplements as promising solutions for mitigating transportation stress. Herbal supplements contain bioactive compounds with anti-inflammatory, antioxidant, and adaptogenic properties, making them effective in reducing oxidative damage and stabilizing cortisol levels. Examples include ashwagandha (*Withania somnifera*) [9], turmeric (*Curcuma longa*) [10], aloe vera [11], and many more products that have demonstrated potential in enhancing the immune function of livestock under stress. These natural remedies offer a sustainable, cost-effective alternative to pharmaceutical interventions.

Similarly, prebiotic supplements, such as fructooligosaccharides (FOSs) and mannan-oligosaccharides (MOSs), have shown promise in improving gut health and overall resilience during stress [12]. By promoting the growth of beneficial gut bacteria, prebiotics stabilize the gut microbiome, enhance immune responses, and improve nutrient absorption during stress [8]. These benefits are particularly relevant for ruminants, as their digestive efficiency and immunity are closely tied to their gut health.

Transportation stress in animals is a concern as it affects animal welfare. Supplementation with anti-stressor compounds like prebiotics, probiotics, and herbal preparations is currently used in research studies and further in routine veterinary therapeutics. The efficacy of these supplements still needs to be tested through research and clinical studies as these supplements are claimed to have long-term advantages with no residues in animals’ bodies. Despite the potential benefits of herbal and prebiotic supplements, research on their use to mitigate transportation stress in sheep remains limited. Existing studies primarily focus on either interventions alone or on other livestock species, leaving a significant gap in our understanding of their individual and comparative effects. Moreover, there is insufficient evidence on their application under harsh climatic conditions, such as the high temperatures characteristic of summer transportation in arid regions.

This study aims to address these gaps by evaluating the impact of herbal and prebiotic oral supplements on transportation stress in Kenguri sheep. The research focuses on a comprehensive assessment of physiological parameters, blood variables, and molecular responses to establish the efficacy of these interventions. It is hypothesized that herbal and prebiotic supplements will mitigate the adverse effects of transportation stress compared to the control group, with distinct benefits in physiological stability, immune response, and metabolic regulation. The findings are expected to contribute valuable insights into sustainable livestock management practices, providing practical and low-cost solutions that align with both economic and welfare objectives.

## 2. Materials and Methods

### 2.1. Study Location and Ethical Approval

The study was conducted at two locations: Bannur town, Karnataka, India (654 m above mean sea level; 76.86° E longitude, 12.33° N latitude), where the sheep were loaded, and the Centre for Climate Resilient Animal Adaptation Studies (CCRAAS), ICAR-NIANP, Bengaluru, India (920 m above mean sea level; 77°38′ E longitude, 12°58′ N latitude), where post-load observations were recorded. Ethical approval was obtained from the Institutional Animal Ethics Committee (IAEC), constituted under article no. 13 of the CPCSEA rules, Government of India (Reference No. V-11011(13)/14/2021-CPCSEA-DADF). Prophylactic measures were implemented to ensure the health of all experimental animals prior to the study.

### 2.2. Experimental Animals and Grouping

Thirty healthy female Kenguri sheep, aged 8 months and weighing 10–15 kg, were selected for this study. The animals were randomly divided into three groups (*n* = 10 per group) based on their body weight: a control group (CKS) that received no supplementation, an herbal supplementation group (HKS) that was administered 10 g of Herbal Transcare powder dissolved in 10 mL of sterile water, and a prebiotic supplementation group (PKS) that received 10 g of Prebiotic Transcare dissolved in 10 mL of sterile water. The supplements were administered orally 45–60 min prior to transportation. Table 1 describes the composition of the prebiotic supplement and herbal supplement.

### 2.3. Transportation Protocol

Animals were fasted with restricted access to food and water 7 h prior to transportation to simulate real-world transport conditions. They were transported in an open-roof pick-up van with a stocking density of 0.6 m^2^ per 100 kg of animal weight, in accordance with established animal welfare guidelines [13,14]. Wood shavings were dispersed on the truck floor to ensure stable footing during transit. The vehicle traveled 230 km over 7 h on paved roads, maintaining an average speed of 40–50 km/h. Environmental conditions during transport ranged from a minimum temperature of 32.5 °C to a maximum of 34.9 °C. None of the sheep were in estrus during transport.

### 2.4. Weather Variables and THI Calculation

Environmental parameters, including maximum and minimum temperatures, dry and wet bulb temperatures, ambient temperature, and relative humidity, were carefully recorded using standard procedures. The minimum and maximum temperature, ambient temperature, and relative humidity were recorded using digital thermo hygrometer. The dry and wet bulb temperatures were recorded using dry and wet bulb thermometers. All these weather variables were recorded at one point in the middle of the transport route at 14:00 h. These variables were then integrated into the McDowell [15] formula to calculate the THI, providing a comprehensive assessment of the climatic challenges faced by the sheep during the study. This index enabled a detailed understanding of the thermal environment and its potential effects on animal welfare and physiological responses.

### 2.5. Physiological and Hematological Measurements

Post-transport physiological parameters were assessed to evaluate the impact of transportation stress and supplementation. Respiration rate (RR) was determined by observing flank movements at the paralumbar fossa and expressed as breaths per minute, while pulse rate (PR) was recorded by palpating the femoral artery and counting the number of pulses per minute. Rectal temperature (RT) was measured using a clinical digital thermometer placed in contact with the rectal mucosa for 60 s, with results expressed in degrees Celsius (°C) [16]. Skin temperature (ST) was documented using a non-contact infrared thermometer (B.S.K. Technologies, Hyderabad, India) held at a distance of 5–15 cm from the animal, with results expressed in degrees Celsius (°C). Blood samples were collected from the external jugular vein using sterile 20-gauge needles in heparinized tubes (20 IU/mL) and analyzed for hematological parameters, including red blood cell count (RBC), hemoglobin concentration (HGB), hematocrit (HCT), mean corpuscular volume (MCV), mean corpuscular hemoglobin (MCH), mean corpuscular hemoglobin concentration (MCHC), red cell distribution width (RDW), and white blood cell count (WBC), using an automated hematological analyzer (Mindray BC 2800).

Following collection, the blood samples underwent centrifugation at 3500 rpm for 20 min to separate the plasma. The isolated plasma was then stored at −20 °C to facilitate subsequent analysis of plasma glucose (post-load). Glucose levels were determined using Span diagnostic kits, India, following the standard method and utilizing a BioSpectrophotometer (Eppendorf, Hamburg, Germany).

### 2.6. PBMC Isolation and Gene Expression Analysis

Peripheral blood mononuclear cells (PBMCs) were isolated from blood samples using RBC lysis buffer. The protocol used for total RNA isolation, cDNA synthesis, primer sequences for amplifying the gene of interest, and relative expression of the genes by real-time qPCR were established according to the protocol described by [3], with primer sequences listed in Table 2. Gene expression was normalized against GAPDH as the internal control, and relative expression was calculated using the 2^-ΔΔCT method [17].

### 2.7. Statistical Analysis

Data were analyzed using the Mixed Model procedure in SPSS software version 20.0 (SPSS Inc., Chicago, IL, USA). The treatment factors were considered as fixed effects and animal factors were considered as random effects. The significance of each effect on physiological and hematological parameters was assessed by least squares analysis (LSD), while Tukey’s test was employed for pairwise comparisons in gene expression analysis following one-way ANOVA [18]. Results are expressed as mean ± SEM, and statistical significance was set at *p* < 0.05.

## 3. Results

### 3.1. Temperature Humidity Index

The THI recorded both before and during transportation is described in Table 3. The THI was in a comfortable range both inside and outside the vehicle before transportation. However, the THI was extremely stressful inside the vehicle and moderately stressful outside the vehicle during transportation.

### 3.2. Physiological Responses

The physiological responses of the Kenguri sheep to transportation stress, along with the effects of prebiotic and herbal supplementation, are detailed in Table 4. The CKS exhibited a significantly elevated RR and PR compared to the PKS and HKS. The RR in CKS was 38.7 ± 1.38 breaths/min, which was significantly reduced to 29.6 ± 0.80 in the PKS and 31.8 ± 1.10 in the HKS (*p* < 0.01). Similarly, the PR was higher in the CKS (75.6 ± 2.61 pulses/min) and significantly decreased in the PKS (64.8 ± 2.09 pulses/min) and the HKS (66.8 ± 2.04 pulses/min) (*p* < 0.05).

Conversely, the RT, skin temperature head (STH), and skin temperature shoulder (STS) values did not differ significantly across the groups (*p* > 0.05). The RT values remained comparable among the groups (CKS: 39.44 ± 0.20 °C; PKS: 40.00 ± 0.26 °C; HKS: 40.00 ± 0.26 °C), indicating no influence of supplementation on these physiological parameters during transportation stress. Similarly, the STH and STS values showed no significant variation, as summarized in Table 4.

### 3.3. Hematological and Biochemical Parameters

The effects of transportation stress and supplementation on hematological and biochemical variables are presented in Table 5. Among the measured parameters, HGB and HCT levels showed significant differences among the groups. The CKS exhibited higher HGB levels (9.18 ± 0.33 g/dL) compared to the PKS (7.87 ± 0.32 g/dL) and the HKS (7.97 ± 0.31 g/dL) (*p* < 0.05). Similarly, the HCT was significantly elevated in the CKS (28.1 ± 0.87%) relative to the PKS (24.2 ± 0.94%) and the HKS (24.7 ± 0.80%) (*p* < 0.05). Further, the plasma glucose level was significantly higher (*p* > 0.01) in the HKS group as compared to the CKS group. However, the plasma glucose in the PKS group was not significant in either the CKS or HKS groups. Other hematological variables, including the RBC count, MCV, MCH, MCHC, RDW, and WBC, did not show significant differences among the CKS, PKS, and HKS groups (*p* > 0.05). These results suggest that while the HGB, HCT, and glucose levels were influenced by supplementation, the other hematological markers remained unaffected by the prebiotic or herbal interventions.

### 3.4. Molecular Responses

The effects of supplementation on molecular chaperone responses in Kenguri sheep are summarized in Table 6. A gene expression analysis revealed no statistically significant differences in the fold changes of *HSF1*, *HSP70*, *HSP90*, or *HSP27* among the CKS, PKS, and HKS groups (*p* > 0.05). However, non-significant trends were observed for *HSP70*, with the highest expression noted in the HKS group (3.12 ± 1.38), followed by the PKS (1.93 ± 0.34) and the CKS (1.08 ± 0.11). Similarly, *HSP27* expression was slightly elevated in the PKS (1.23 ± 0.10) compared to the HKS (1.14 ± 0.11) and the CKS (1.06 ± 0.09), but these differences were not statistically significant. These findings indicate that while transportation stress induced some molecular responses, supplementation with prebiotic or herbal products did not significantly alter the expression levels of the molecular chaperones evaluated.

## 4. Discussion

As per McDowell’s [15] model, THI values of 72 or less are considered comfortable, THI values between 75 and 78 are considered stressful, and THI values above 78 are considered extremely distressful. Thus, the THI obtained both inside (69.1) and outside (71.2) the vehicle clearly indicated that the THI was in the comfortable range before transportation. Further, the THI value of 78.6 obtained inside the vehicle during transportation clearly demonstrated the extremely severe heat stress experienced by these animals. The THI of 76.2 recorded during transportation outside the vehicle falls in the category of moderate heat stress. This indicates that the heat stress was of a higher magnitude inside the vehicle than outside during transportation.

While it is true that prebiotics such as inulin, mannan-oligosaccharides (MOSs), fructooligosaccharides (FOSs), and galactooligosaccharides (GOSs) are susceptible to partial fermentation by rumen microbiota, studies indicate that a significant proportion can escape complete ruminal degradation and reach the hindgut, where they exert their functional effects. The degree of degradation depends on the chain length, rumen passage rate, and microbial composition [19,20]. It has been reported that short-chain oligosaccharides, particularly GOSs and FOSs, may resist complete fermentation in the rumen and retain functionality in the lower gut [21].

Physiological stress responses, such as changes in cortisol levels, oxidative stress markers, and inflammatory responses, can occur rapidly after exposure to a stressor or an intervention. Several studies have demonstrated that oral supplementation with nutritional supplements (bioactives or adaptogens) can lead to measurable changes in antioxidant enzyme activities and stress markers within 4–8 h of administration [22,23]. Therefore, it is plausible that a single dose can exert detectable physiological effects within an 8 h window, especially in the context of acute stress, such as during transportation.

Prebiotics and herbal supplements have been shown to influence gut microbiota composition, increase the production of short-chain fatty acids (SCFAs), and stimulate gut-associated lymphoid tissue (GALT), all of which contribute to the systemic modulation of immunity and oxidative stress responses. These bioactive compounds in nutritional supplements enhance endogenous antioxidant defense mechanisms through the upregulation of enzymes like superoxide dismutase (SOD), catalase (CAT), and glutathione peroxidase (GPx) [24,25]. Additionally, herbal adaptogens can exert rapid neuroendocrine modulation via the HPA axis, contributing to reduced oxidative damage and improved cellular homeostasis.

Transportation stress significantly impacts the physiological parameters of livestock, including RR, PR, RT, STH, and STS. These parameters are interconnected, reflecting the systemic and thermoregulatory challenges animals face during transit. In this study, both prebiotic and herbal supplementation effectively reduced RR and PR compared to the CKS, though no significant differences were observed in RT and skin temperature metrics across the groups.

The elevated RR (38.7 ± 1.38 breaths/min) and PR (75.6 ± 2.61 beats/min) values observed in the CKS group reflect heightened sympathetic nervous system activity, a hallmark of acute stress. These values indicate increased metabolic and oxygen demands triggered by transportation stress, consistent with previous findings in goats under similar conditions [3]. The significant reductions in RR (29.6 ± 0.80 breaths/min) and PR (64.8 ± 2.09 beats/min) in the PKS group and RR (31.8 ± 1.10 breaths/min) and PR (66.8 ± 2.04 beats/min) in the HKS group highlight the efficacy of the prebiotic and herbal supplementation in mitigating these stress-induced physiological responses.

Prebiotics, such as FOSs and inulin, are known to influence the gut–brain axis by modulating microbial populations and promoting short-chain fatty acid (SCFA) production. SCFAs, including acetate, propionate, and butyrate, have demonstrated blood-pressure-lowering effects [26], which may contribute to the observed reductions in PR in the PKS group. By improving autonomic nervous system regulation [27], SCFAs also stabilize RR, further supporting the systemic effects of prebiotic supplementation. Similarly, β-glucans in the PKS group are recognized for their immunomodulatory properties, which reduce systemic inflammation and support autonomic balance [28], contributing to the observed reductions in RR and PR.

*Withania somnifera* has been shown to significantly reduce damage to the heart caused by ischemia, primarily through its anti-apoptotic properties and its ability to restore oxidative balance [29]. This cardio-protective effect of *Withania somnifera* upregulates the mitochondrial anti-apoptotic pathway through increased AMPK phosphorylation and the Bcl-2/Bax ratio [30]. By maintaining cellular energy homeostasis and regulating glucose and lipid metabolism, AMPK activation regulates glucose and restores energy balance [31], therefore improving cardiovascular and respiratory efficiency under stress. These mechanisms may explain the observed reductions in the PR in the HKS group, as improved heart function reduces the cardiovascular workload required to maintain homeostasis during stress. In addition, *Tinospora cordifolia* is known to enhance parasympathetic activity, reducing sympathetic over-activation and stabilizing autonomic function [32]. *Morinda citrifolia*, with its ability to promote nitric oxide synthesis, likely contributed to vascular relaxation [33] and reduced peripheral resistance [33], further supporting the observed reduction in PR. The antioxidant properties of *Aloe barbadensis* enhanced circulatory efficiency, indirectly reducing the metabolic and respiratory demands [34], which may account for the lower RR in the HKS group.

Unlike RR and PR, no significant differences in the RT or skin temperatures were observed across the groups. The mean RT values (CKS: 39.44 ± 0.20 °C, PKS: 40.00 ± 0.26 °C, HKS: 40.00 ± 0.26 °C) indicate that transportation stress did not lead to substantial hyperthermia in any of the groups. The consistent RT values, in spite of these animals being subjected to extreme heat stress as reflected by the THI results, may be attributed to the indigenous nature of Kenguri sheep, and thus, the animals were able to maintain their body temperature through physiological adjustments that contributed to heat dissipation. Similarly, skin temperatures for the head (STH: 31.4 ± 0.41 °C in CKS; 31.5 ± 0.38°C in PKS; 30.9 ± 0.45 °C in HKS) and shoulder (STS: 30.3 ± 0.39 °C in CKS; 31.2 ± 0.40°C in PKS; 31.6 ± 0.26 °C in HKS) did not vary significantly, indicating that peripheral heat dissipation mechanisms remained stable across the groups. The lack of significant variation in RT and skin temperatures across the groups may reflect the limited direct influence of supplementation on thermoregulation. Instead, the observed reductions in RR and PR in the PKS and HKS groups may have indirectly supported thermal homeostasis by reducing metabolic heat production and enhancing circulatory efficiency.

The significant reductions in RR and PR in the PKS and HKS groups suggest a coordinated improvement in systemic stress responses, with prebiotic supplementation targeting gut-mediated metabolic pathways and herbal supplementation modulating hormonal and antioxidant defenses. The blood-pressure-lowering effects of SCFAs, including acetate, propionate, and butyrate, may further explain the reductions in PR, as a lower blood pressure often corresponds with stabilized heart rate under stress conditions [18]. The consistent RT and skin temperature values indicate that while supplementation had limited direct effects on thermoregulation, the improved physiological stability in the supplemented groups may have supported peripheral heat balance indirectly.

Transportation stress significantly affects hematological parameters in livestock. This study observed changes in key blood parameters of Kenguri sheep, with prebiotic and herbal supplementation showing potential to mitigate these effects compared to the control group. The CKS group displayed higher HGB levels (9.18 ± 0.33 g/dL) and HCT percentages (28.1 ± 0.87%) compared to the PKS and HKS groups, with statistically significant differences (*p* = 0.033 for HGB and *p* = 0.030 for HCT). Elevated HGB and HCT levels in the CKS group may indicate hemoconcentration, a common response to acute stress in which fluid shifts from the intravascular to extravascular compartments occur, leading to an increased red blood cell concentration. This physiological response has been observed in the control group of cattle subjected to transportation stress, as compared to a supplemented group, in which stress-induced dehydration led to hemoconcentration [14].

In contrast, the PKS and HKS groups exhibited lower HGB and HCT values, suggesting that the supplementation may have helped stabilize these parameters. The reduction in hemoconcentration in the supplemented groups could indicate a more balanced physiological response to transportation stress, likely due to improved overall metabolic and stress resilience. Prebiotic supplementation may enhance nutrient absorption and systemic stability, while herbal components with antioxidant properties may support vascular health and reduce oxidative stress, contributing to more stable hematological profiles. Although RBC counts, MCV, MCH, MCHC, RDW, and WBC counts did not show statistically significant differences between the groups, the observed trends are consistent with earlier studies [3]. The lack of significant differences in some hematological parameters may be attributed to the duration and conditions of transportation, as well as the timing of blood sample collection relative to the animals’ exposure to stress. It is also possible that the supplementation protocols require optimization in dosage or duration to elicit more pronounced effects on these parameters.

This study assessed the impact of transportation stress on glucose levels in Kenguri sheep and the potential mitigation effects of herbal and prebiotic supplementation. The significant difference in plasma glucose level among the CKS and PKS groups offer important insights into how these interventions influence metabolic responses under stress. The control group exhibited the lowest mean glucose levels (63.5 mg/dL) while the highest level was established in the HKS group (80.0 mg/dL) among the three groups, reflecting the mitigating effects of the herbal supplement for transportation stress. Stress activates the hypothalamic–pituitary–adrenal (HPA) axis, leading to increased cortisol secretion, gluconeogenesis, and hyperglycemia [35]. However, the combination of increased metabolic demand (evidenced by an elevated respiration rate and pulse rate in the CKS) and inadequate metabolic buffering likely resulted in rapid glucose depletion. Similar patterns of glucose reduction under stress have been observed in cattle, in which energy demands exceed glucose production, leading to transient hypoglycemia [36]. The absence of supplementation in the CKS group left the animals vulnerable to systemic dysregulation, highlighting the need for nutritional interventions to support glucose metabolism and physiological stability during stress.

The HKS group displayed the highest glucose levels (80.0 ± 5.07 mg/dL), suggesting that herbal supplementation influenced glucose metabolism during transportation. The adaptogenic properties of herbal components such as *Withania somnifera* and *Ocimum sanctum* are known to modulate the HPA axis, reducing cortisol levels under stress while enhancing acute glucose mobilization for energy [37]. This adaptive glucose mobilization ensures that sufficient energy is available to meet the demands of transportation stress. In addition, components such as *Tinospora cordifolia* and *Aloe barbadensis* are recognized for their antioxidant properties, which can mitigate oxidative stress and inflammation, indirectly supporting glucose homeostasis [34,38]. The elevated glucose levels in the HKS group likely reflect a strategic metabolic adaptation, prioritizing energy availability during acute stress while minimizing oxidative damage and immune suppression.

The PKS group showed intermediate glucose levels (69.6 ± 2.99 mg/dL), indicating a more stable metabolic response compared to the control. Prebiotics such as MOS, FOS, GOS, inulin, and β-glucan are known to promote gut health by enhancing the growth of beneficial bacteria like *Bifidobacteria* and *Lactobacilli* [39]. As mentioned previously, these bacteria produce SCFAs such as acetate and propionate, which improve insulin sensitivity and regulate glucose metabolism [40]. Inulin further stabilizes glucose levels by improving gut–brain signaling pathways [41], ensuring a controlled metabolic response under stress conditions. The PKS group’s glucose levels suggest that prebiotic supplementation helps buffer the metabolic impacts of stress, maintaining glucose homeostasis and preventing both depletion (as observed in the CKS) and excessive mobilization (as seen in the HKS).

The results highlight distinct metabolic responses to stress across the three groups. The control group’s low glucose levels reflect the unbuffered impacts of stress, with increased energy demands and systemic dysregulation leading to rapid glucose depletion. In contrast, the herbal and prebiotic groups displayed improved metabolic resilience, albeit through different mechanisms. The HKS appears to prioritize acute energy mobilization, enhancing glucose availability to meet transportation demands, while the PKS stabilizes glucose levels by promoting gut health and reducing systemic inflammation. The differences between the HKS and PKS suggest that prebiotics provide a more sustained and controlled glucose response, aligning with their role in maintaining gut-derived metabolic stability. Meanwhile, the adaptogenic properties of herbal supplements offer a more dynamic response, ensuring energy availability during periods of acute stress.

The expression of *HSPs* and *HSF1* provides a molecular lens to evaluate the effects of transportation stress and the mitigating roles of prebiotic and herbal supplementation. The trends observed in this study, though not statistically significant, reveal important insights into the cellular stress responses in Kenguri sheep and suggest distinct mechanisms of action for the two supplementation strategies.

In contrast to the CKS, the elevated *HSF1* expression in the PKS group (1.51 ± 0.08) suggests an activation of stress-responsive pathways, consistent with the role of *HSF1* in regulating HSP production [42]. Prebiotics are known for their systemic anti-inflammatory effects and modulation of cellular stress responses through the production of SCFAs. This mechanism likely underpins the observed trends in *HSP70* expression in the PKS group, where the fold change (1.93 ± 0.34) was notably higher than in the control group (1.08 ± 0.11). SCFAs not only stabilize the gut–brain axis but also influence mitochondrial function [43], indirectly supporting protein folding and repair under stress conditions.

The HKS group displayed a distinct pattern, characterized by a significant increase in *HSP70* expression (3.12 ± 1.38). While *Withania somnifera* is known for its anti-inflammatory properties, attributed to its ability to destabilize or downregulate HSP activity involved in regulatory kinase signaling pathways [44], our results reveal a contradictory trend in *HSP70* expression. Unlike *HSP90* and *HSP27*, in which its effects are evident, the response in *HSP70* appears to diverge. Interestingly, while *HSP90* expression remained stable across the groups, a slight decrease was noted in the HKS group (0.81 ± 0.09), potentially reflecting reduced chaperone demand due to the antioxidative effects of herbal supplementation, as mentioned above. Similarly, *HSP27*, which plays a role in inhibiting apoptosis, showed modest increases in both supplemented groups compared to the control. These trends indicate a subtle enhancement of cellular protective mechanisms, further supported by the antioxidant properties of *Tinospora cordifolia* and *Aloe barbadensis*. The interplay between *HSP27* and *HSP70* in the supplemented groups may represent a coordinated effort to maintain cellular integrity during transportation stress.

## 5. Conclusions

This study highlights the significant impact of transportation stress on Kenguri sheep and demonstrates the potential of prebiotic and herbal supplementation to mitigate these effects. The THI obtained from the study indicated that the Kenguri ewes were subjected to heat stress during transportation. The low levels of stress markers such as RR, PR, HBG, and HCT in both the PKS and HKS groups as compared to the CKS group clearly demonstrate the transportation-stress-mitigating potential of both prebiotic and herbal supplements. Further, the herbal supplement also significantly increased the plasma glucose in the HKS group as compared to the CKS group, which reflects the additional benefits of the herbal supplement.

These results provide a scientific basis for incorporating such supplements into livestock management practices, especially in arid and semi-arid regions, where transportation stress is exacerbated by high temperatures. Future research should focus on exploring the long-term effects of these interventions, their mechanistic pathways, and their potential for application across diverse livestock species and transport conditions.

## Figures and Tables

**Table 1 vetsci-12-00442-t001:** (**a**) Composition of prebiotic supplement. (**b**) Composition of herbal supplement.

**(a) Nutrient**	**Composition (100 g)**
Mannan oligosaccharide	15 g
Fructooligosaccharides	30 g
Galacto-oligosaccharide	15 g
Inulin	15 g
β-Glucan	25 g
Dose rate	10 g/Animal/Day
**(b) Nutrient**	**Composition (100 g)**
*Withania somnifera*	15 g
*Ocimum sanctum*	30 g
*Tinospora cordifolia*	20 g
*Morinda citrifolia*	15 g
*Aloe barbadensis*	20 g
Dose rate	10 g/Animal/Day

g Grams.

**Table 2 vetsci-12-00442-t002:** Primers used for relative expression of the targeted genes.

Gene ID		Primer Sequence (5′-3′)	Product Length	Accession No.
*GAPDH*	F R	TCGGAGTGAACGGATTTGGC ACGATGTCCACTTTGCCAGT	76	NM_001190390.1
*HSF-1*	F R	GAAAGTGACCAGCGTGTCCA GTCGGTCAGCAGCTTGGTAA	79	XM_027973370.2
*HSP27*	F R	GAAGCCGGAAAGTCCGAACA CATAGAGGTTTGGCGGGTGA	106	XM_027961472.3
*HSP70*	F R	CCCACCATTGAGGAAGTGGAT CAAACTGACACAGCACAGGAC	102	XM_042236765.2
*HSP90*	F R	ACCTGTCAGACAACGTCCAC TCCATGAGGGCACATTTCTCC	100	XM_060416283.1

F Forward; R Reverse; *GAPDH* Glyceraldehyde-3-phosphate dehydrogenase; *HSF-1* Heat shock transcription factor 1; *HSP27* Heat shock proteins 27; *HSP70* Heat shock protein 70; *HSP90* Heat shock protein 90.

**Table 3 vetsci-12-00442-t003:** Temperature humidity index pre- and post-transport of sheep.

Status of Transport	Inside the Vehicle	Outside the Vehicle
Pre-Transport	69.1	71.2
During Transport	78.6	76.2

**Table 4 vetsci-12-00442-t004:** Effect of prebiotic and herbal supplementation on physiological responses in Kenguri sheep during transportation stress.

Parameter	Group-1 (CKS)	Group-2 (PKS)	Group-3 (HKS)	SEM	*p*-Value
RR (breaths/min)	38.7 ^a^	29.6 ^b^	31.8 ^b^	1.2	0.001
PR (pulses/min)	75.6 ^a^	64.8 ^b^	66.8 ^b^	2.4	0.009
RT (°C)	39.7	40.0	40.0	0.2	0.334
STH (°C)	31.4	31.5	30.9	0.5	0.563
STS (°C)	30.3	31.2	31.6	0.4	0.081

*RR* Respiration Rate; *PR* Pulse Rate; *RT* Rectal Temperature; *STH* Skin Temperature Head; *STS* Skin Temperature Shoulder; °C Degree Celsius; /per; *CKS* Control Kenguri Sheep; *PKS* Prebiotic-Supplemented Kenguri Sheep; *HKS* Herbal-Supplemented Kenguri Sheep; *SEM* Standard Error of Mean. Values are expressed as mean ± SEM. Different superscripts indicate significant differences (*p* < 0.05) between groups.

**Table 5 vetsci-12-00442-t005:** Effect of prebiotic and herbal supplementation on hemato-biochemical variables in Kenguri sheep during transportation stress.

Parameter	Group-1 (CKS)	Group-2 (PKS)	Group-3 (HKS)	SEM	*p*-Value
RBC (10^6^/µL)	11.5	10.1	10.3	0.6	0.204
HGB (g/dL)	9.18 ^a^	7.87 ^b^	7.97 ^b^	0.35	0.023
HCT (%)	28.1 ^a^	24.2 ^b^	24.7 ^b^	0.9	0.021
MCV (fL)	24.9	24.8	24.5	0.9	0.951
MCH (pg)	8.01	7.87	7.80	0.23	0.802
MCHC (g/dL)	32.6	32.1	32.2	0.4	0.605
RDW (%)	17.5	17.3	17.1	0.3	0.570
WBC (10^3^/µL)	10.4	13.8	13.4	1.0	0.055
Glucose (mg/dL)	63.5 ^b^	69.6 ^ab^	80.0 ^a^	3.6	0.012

*RBC* Red Blood Cells; *HGB* Hemoglobin; *HCT* Hematocrit; *MCV* Mean Corpuscular Volume; *MCH* Mean Corpuscular Hemoglobin; *MCHC* Mean Corpuscular Hemoglobin Concentration; *RDW* Red Cell Distribution Width; *WBC* White Blood Cell; *AT* After Transportation; µL Microliter; g/dL Grams Per Deciliter; % Percentage; fL Femtoliter; pg Picogram; mg/dL Milligrams Per Deciliter; *CKS* Control Kenguri Sheep; *PKS* Prebiotic-Supplemented Kenguri Sheep; *HKS* Herbal-Supplemented Kenguri Sheep; *SEM* Standard Error of Mean. Values are expressed as mean ± SEM. Different superscripts indicate significant differences (*p* < 0.05) between groups.

**Table 6 vetsci-12-00442-t006:** Effect of prebiotic and herbal supplementation on molecular chaperone responses in Kenguri sheep during transportation stress.

Gene	Group	Fold Change	SEM	*p*-Value
*HSF1*	CKS	1.08	0.18	0.410
	PKS	1.51		
	HKS	1.24		
*HSP90*	CKS	1.08	0.1	0.191
	PKS	1.04		
	HKS	0.81		
*HSP70*	CKS	1.08	0.61	0.227
	PKS	1.93		
	HKS	3.12		
*HSP27*	CKS	1.06	0.1	0.525
	PKS	1.23		
	HKS	1.14		

*HSF 1* Heat Shock Factor 1; *HSP90* Heat Shock Protein 90; *HSP70* Heat Shock Protein 70; *HSP27* Heat Shock Protein 27; CKS Control Kenguri Sheep; PKS Prebiotic-Supplemented Kenguri Sheep; HKS Herbal-Supplemented Kenguri Sheep; *SEM* Standard Error of Mean.

## Data Availability

All relevant data are presented within the paper.

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
