# Peer review of "The Role of Prebiotic and Herbal Supplementation in Enhancing Welfare and Resilience of Kenguri Sheep Subjected to Transportation Stress"

_vetsci, 2025, doi:10.3390/vetsci12050442_

Round 1

Reviewer 1 Report

Comments and Suggestions for Authors

This study investigates the effects of prebiotic and herbal supplementation on physiological, hematological, and molecular stress responses in Kenguri sheep during transportation. The research is well-designed, with clear objectives and a structured methodology, and it contributes valuable insights into stress mitigation strategies for livestock. However, some areas require clarification or improvement to enhance the manuscript’s rigor, readability, and impact.

Line 114, the reason for the dose of the HKS and PKS?

The mean and SEM are presented with two decimal places in the table.

Line 28: “trends suggested higher HSP70 expression in the HKS group” What is the basis or standard for this conclusion? There was no statistical significance in Table 3. Please explain and supplement the relevant content in the article. Excessive extrapolation should be avoided.

Animal discomfort is common after long distance transportation, so whether its subsequent recovery after the use of prebiotic and herbal supplementation should be continued to be observed, such as changes in feed intake, animal mental status, disease occurrence, and comparison of meat quality.

The conclusion should be clear and concise.

Author Response

Reviewer 1

Comment 1: This study investigates the effects of prebiotic and herbal supplementation on physiological, hematological, and molecular stress responses in Kenguri sheep during transportation. The research is well-designed, with clear objectives and a structured methodology, and it contributes valuable insights into stress mitigation strategies for livestock. However, some areas require clarification or improvement to enhance the manuscript’s rigor, readability, and impact.

Response: We thank the reviewer for the encouraging comments on our manuscript. We also addressed all the comments raised by this reviewer in appropriate places in the revised manuscript.

Comment 2: Line 114, the reason for the dose of the HKS and PKS?

Response: The dosage of 10g per animal for both the herbal and prebiotic supplements was determined based on a comprehensive review of relevant literature involving small ruminants. Studies utilizing similar herbal and prebiotic formulations for stress alleviation in sheep have reported effective results at comparable dosage levels. These references were used to select a physiologically appropriate and safe dose that could be expected to exert a beneficial effect without causing adverse reactions. The chosen dose also aligns with recommended practices for small ruminants in previous research, ensuring consistency and scientific justification for its use in the current study.  

Comment 3: The mean and SEM are presented with two decimal places in the table.

Response: This issue was fixed for mean and SEM. This can be found in tables 4 and table 5 of the revised manuscript.

Comment 4: Line 28: “trends suggested higher HSP70 expression in the HKS group” What is the basis or standard for this conclusion? There was no statistical significance in Table 3. Please explain and supplement the relevant content in the article. Excessive extrapolation should be avoided.

Response: We agree with the reviewer on this particular point and we removed this sentence from the abstract in the revised manuscript. Also care was taken to avoid such extrapolation in the manuscript.

Comment 5: Animal discomfort is common after long distance transportation, so whether its subsequent recovery after the use of prebiotic and herbal supplementation should be continued to be observed, such as changes in feed intake, animal mental status, disease occurrence, and comparison of meat quality.

Response: We completely agree with the reviewer and appreciate the reviewer for this valuable point. Unfortunately we could not do these variables in this study. We will keep this in mind in our subsequent studies.

Comment 6: The conclusion should be clear and concise.

Response: This particular point raised by the reviewer was considered and the conclusion was precisely reduced reflecting only the results obtained from the study. The changes can be found in lines 469-475 & 477-478 of the revised manuscript.

Reviewer 2 Report

Comments and Suggestions for Authors

Authors have conducted an interesting study. My decision is based on the follow comments:

  1. The main concern is about similar publications from first author. In M&M section authors indicated that a “Transcare powder” were evaluated. The same product with the same methodology and conceptualization was evaluated in sheep and published by Sejian, V., Devaraj, C., Shashank, C.G. et al. Mitigating transportation stress in Bannur sheep: exploring the utility of innovative antioxidant supplementation in a hot-dry tropical climate. Trop Anim Health Prod 57, 115 (2025). https://doi.org/10.1007/s11250-025-04364-0.
  2. Several questions I have about action mechanism of prebiotics and probiotics. Firstly, how do you guarantee that probiotics (inulin, mannan, fructo and galactooligosaccharides) and prebiotics were not degraded by rumen microorganisms? How it is possible that a single oral doses can modified physiological changes at 8 h post oral administration? How did improve gut-derived systemic modulation and antioxidant and adaptogenic properties to enhance cellular defenses? It does not a physiological make sense.
  3. Statistical analysis should be reconsidered. In this case, treatments were offered oral via, and then a mixed model instead of lineal model to data analysis. It is due to sheep is considered a random and treatments fixed components in the model.
  4. There are several studies with prebiotic and probiotic effects in ruminants.
  5.  

Author Response

Reviewer 2

Comment 1: Authors have conducted an interesting study. My decision is based on the follow comments:

Response: We thank the reviewer for the encouraging comment. We have addressed all the queries of this reviewer.

Comment 2: The main concern is about similar publications from first author. In M&M section authors indicated that a “Transcare powder” were evaluated. The same product with the same methodology and conceptualization was evaluated in sheep and published by Sejian, V., Devaraj, C., Shashank, C.G. et al. Mitigating transportation stress in Bannur sheep: exploring the utility of innovative antioxidant supplementation in a hot-dry tropical climate. Trop Anim Health Prod 57, 115 (2025). https://doi.org/10.1007/s11250-025-04364-0.

Response: We would like to clarify that the mentioned paper uses anitoxidants supplements– betaine, methionine, copper, cobalt, selenium, Vit, C, Vit E, dextrose, sodium, chloride, & sodium bicarbonate. In the current study we used two different supplements –

  1. Prebiotic - Mannan oligosaccharide, Fructooligosaccharides, Galacto- oligosaccharide, Inulin & β-Glucan
  2. Herbal supplements - Withania somnifera, Ocimum sanctum, Tinospora cordifolia, Morinda citrifolia & Aloe barbadensis

Therefore this is totally a different study using prebiotic transcare and herbal transcare to relieve transportation stress in Kenguri breed of sheep. 

Comment 3: How do you guarantee that probiotics (inulin, mannan, fructo- and galacto-oligosaccharides) and prebiotics were not degraded by rumen microorganisms?

Response: While it is true that prebiotics such as inulin, mannan-oligosaccharides (MOS), fructooligosaccharides (FOS), and galactooligosaccharides (GOS) are susceptible to partial fermentation by rumen microbiota, studies indicate that a significant proportion can escape complete ruminal degradation and reach the hindgut, where they exert their functional effects. The degree of degradation depends on chain length, rumen passage rate, and microbial composition. It has been reported that short-chain oligosaccharides, particularly GOS and FOS, may resist complete fermentation in the rumen and retain functionality in the lower gut. This information was included in the discussion part of the revised manuscript in lines 276-283.

Comment 4: How is it possible that a single oral dose can modify physiological changes at 8 h post oral administration?

Response: Physiological stress responses, such as changes in cortisol levels, oxidative stress markers, and inflammatory responses, can occur rapidly after exposure to a stressor or an intervention. Several studies have demonstrated that oral supplementation with nutritional supplements (bioactives or adaptogens) can lead to measurable changes in antioxidant enzyme activities and stress markers within 4–8 hours of administration. Therefore, it is plausible that a single dose can exert detectable physiological effects within an 8-hour window, especially in the context of acute stress such as transportation. This information was included in the discussion part of the revised manuscript in lines 284-290.

Comment 5: How did it improve gut-derived systemic modulation and antioxidant and adaptogenic properties to enhance cellular defences?

Response: Prebiotics and herbal supplements have been shown to influence gut microbiota composition, increase the production of short-chain fatty acids (SCFAs), and stimulate gut-associated lymphoid tissue (GALT), all of which contribute to systemic modulation of immunity and oxidative stress responses. These bioactive compounds in the nutritional supplement enhance endogenous antioxidant defense mechanisms through upregulation of enzymes like superoxide dismutase (SOD), catalase (CAT), and glutathione peroxidase (GPx). Additionally, herbal adaptogens can exert rapid neuroendocrine modulation via the HPA axis, contributing to reduced oxidative damage and improved cellular homeostasis. This information was included in the discussion part of the revised manuscript in lines 291-299.

Comment 4: Statistical analysis should be reconsidered. In this case, treatments were offered oral via, and then a mixed model instead of lineal model to data analysis. It is due to sheep is considered a random and treatments fixed components in the model.

Response: We thank the reviewer for giving your suggestion to improve the analysis. We have re-analysed the data upon changing the statistical model. As per the reviewer's suggestion we have used mixed model wherein the treatment factors were considered as fixed effects and animal as random effect. The significant effect of each effect on physiological and hematological parameters was assessed by least square analysis (LSD). The changes made are reflected in the materials and methods in lines 190-193 and table 4 and table 5 of the revised manuscript.

Comment 5: There are several studies with prebiotic and probiotic effects in ruminants.

Response: We agree to this point of reviewer. However, we would clarify that this is a first of its kind attempt to use the prebiotic and probiotic feed additives for relieving transportation stress and comparing the results with another group of sheep fed herbal supplements for relieving transportation stress.

Reviewer 3 Report

Comments and Suggestions for Authors

I present my considerations about the manuscript, making the descriptions as bellow:

Lines 16 and 17: the sentence “This study investigated the effects of prebiotic and herbal supplementation on physiological, hematological, and molecular responses in Kenguri sheep subjected to transportation stress” is very similar to the prior sentence (lines 14 and 15) “A study was conducted to assess the efficacy of prebiotic and herbal supplements to ensure welfare and resilience in Kenguri sheep subjected to transportation stress.” that is very similar to the title of the manuscript. So, I suggest that this part of the writting be improved by the authors.

Lines 33 and 34: four Keywords must be replaced, since They are presented in the title of the manuscript:  Kenguri sheep; Transportation stress; Prebiotic supplementation; Herbal 33 supplementation

Lines 112 and 113: “Thirty healthy female Kenguri sheep, aged 8–10 months and weighing 10–15 kg, were selected for this study.” In recent studies, the mean body weight of female Kenguri sheep from Karnataka – India was 20.05 kg (animals aged from 6-12 months). How do the authors justify the light body weight of the animals used in the experiment?

Lines 391, 401, 402, 461, 465, 485, 486, 497 and 523: put the words Withania somnifera, Tinospora cordifolia, Aloe barbadensis, Curcuma longa, Aloe vera, in vitro (2x), in vivo, Morinda citrifolia, Withania somnifera in italics.

References starting from line 435: Standardize the way in which journal´s titles are presented, as some of them are abbreviated.

The main question is not well addressed by the research. It is the effects of the use of prebiotic and herbal supplements on welfare and resilience of female sheeps subjected to stress during transportation. What kind of stress were the animals subjected to during transport? Theoretically, thermal heat stress.

However, although the authors stated that “Environmental parameters, including maximum and minimum temperatures, dry and wet bulb temperatures, ambient temperature, and relative humidity, were carefully recorded using standard procedures. These variables were then integrated into the McDowell [15], formula to calculate the THI, providing a comprehensive assessment of the climatic challenges faced by the sheep during the study.” (Lines 134 to 138), temperature humidity index (THI) was not presented by the authors and was not used in the discussion of the results. Only ambient temperature ranging from 32,5oC to 34,9oC.

It is no coincidence that the authors made the following statement in the discussion of the results: “The consistent RT values may suggest that the ambient temperature during transport and ventilation conditions in the vehicle were sufficient to prevent heat stress.” (Lines 282 to 284). Perhaps there was a mistake, as the animals were subjected to heat stress. However, at the intensity of the heat stress, the animals were able to maintain their body temperature through physiological adjustments that contributed to heat dissipation. Just look at what was written in lines 248 and 249: “These values indicate increased metabolic and oxygen demands triggered by transportation stress.” The animals were able to regulate their body temperature by increasing their respiration rate and pulse rate.

And the authors concluded that: “The findings highlight the practicality of prebiotic and herbal supplements as non-invasive, cost-effective strategies to improve animal welfare and productivity during transportation, particularly in challenging environmental conditions. These results provide a scientific basis for incorporating such supplements into livestock management practices, especially in arid and semi-arid regions where stressors are exacerbated by high temperatures.”(Lines 413 to 418). After all, were the animals subjected to heat stress or not? The answer must be well-founded in the characterization of the THI in which the animals were transported.

Therefore, although the researchers obtained good results with the research, the manuscript needs to be improved in the aspects mentioned.

Author Response

Reviewer 3

Comment 1: Lines 16 and 17: the sentence “This study investigated the effects of prebiotic and herbal supplementation on physiological, hematological, and molecular responses in Kenguri sheep subjected to transportation stress” is very similar to the prior sentence (lines 14 and 15) “A study was conducted to assess the efficacy of prebiotic and herbal supplements to ensure welfare and resilience in Kenguri sheep subjected to transportation stress.” that is very similar to the title of the manuscript. So, I suggest that this part of the writing be improved by the authors.

Response: We thank the reviewer for critically pointing out this. Accordingly, the prior sentence in abstract was removed and the sentence was modified to make it different than the title. The changes are reflected in lines 14-16 in the abstract of the revised manuscript.

Comment 2: Lines 33 and 34: four Keywords must be replaced, since they are presented in the title of the manuscript:  Kenguri sheep; Transportation stress; Prebiotic supplementation; Herbal 33 supplementation

Response: We rectified this mistake in the revised manuscript by replacing four keywords. The change was reflected in line 34 of the revised manuscript.

Comment 3: Lines 112 and 113: “Thirty healthy female Kenguri sheep, aged 8–10 months and weighing 10–15 kg, were selected for this study.” In recent studies, the mean body weight of female Kenguri sheep from Karnataka – India was 20.05 kg (animals aged from 6-12 months). How do the authors justify the light body weight of the animals used in the experiment?

Response: We appreciate the reviewer for this critical comment. We would like to clarify that 12 months body weight averages 20 kg in ewes of this breed. In the current study we predominantly used 8 months old to range between 10-15 kgs. Therefore we corrected the age 8 months old rather than 8-10 months old. This change was reflected in the revised manuscript in line 126.

Comment 4: Lines 391, 401, 402, 461, 465, 485, 486, 497 and 523: put the words Withania somniferaTinospora cordifoliaAloe barbadensisCurcuma longaAloe verain vitro (2x), in vivoMorinda citrifoliaWithania somnifera in italics.

Response: These mistakes were rectified by indicating all these words in italics in the revised manuscript. The changes can be found in lines 453, 463-464, 522, 564-565, 572, 576, 578 and 602 of the revised manuscript.

Comment 5: References starting from line 435: Standardize the way in which journal´s titles are presented, as some of them are abbreviated.

Response: We thank the reviewer for this critical observation. The journal format is that the journal names have to be abbreviated. Accordingly, all journal names in the entire reference section were abbreviated in the revised manuscript.

Comment 6: The main question is not well addressed by the research. It is the effects of the use of prebiotic and herbal supplements on welfare and resilience of female sheeps subjected to stress during transportation. What kind of stress were the animals subjected to during transport? Theoretically, thermal heat stress.

Response: We agree to this particular point raised by the reviewers. With the inclusion of THI result and discussion we believe we have addressed this point by establishing the fact that these animals were subjected to extreme heat stress during transportation. The changes are reflected in lines 198-203 and 267-275 of the revised manuscript.

Comment 7: However, although the authors stated that “Environmental parameters, including maximum and minimum temperatures, dry and wet bulb temperatures, ambient temperature, and relative humidity, were carefully recorded using standard procedures. These variables were then integrated into the McDowell [15], formula to calculate the THI, providing a comprehensive assessment of the climatic challenges faced by the sheep during the study.” (Lines 134 to 138), temperature humidity index (THI) was not presented by the authors and was not used in the discussion of the results. Only ambient temperature ranging from 32,5oC to 34,9oC.

Response: We appreciate the reviewer for this valuable suggestion. Accordingly, a new table indicating the THI both before and during transportation was included in the revised manuscript. Further THI based result and discussion was also included in the revised manuscript. The changes are reflected in lines 198-203; 267-275 and table 3 of the revised manuscript.

Comment 8: It is no coincidence that the authors made the following statement in the discussion of the results: “The consistent RT values may suggest that the ambient temperature during transport and ventilation conditions in the vehicle were sufficient to prevent heat stress.” (Lines 282 to 284). Perhaps there was a mistake, as the animals were subjected to heat stress. However, at the intensity of the heat stress, the animals were able to maintain their body temperature through physiological adjustments that contributed to heat dissipation. Just look at what was written in lines 248 and 249: “These values indicate increased metabolic and oxygen demands triggered by transportation stress.” The animals were able to regulate their body temperature by increasing their respiration rate and pulse rate.

Response: We agree it was a mistake to give inconsistent explanation on the physiological responses. The THI results indicated extremely severe heat stress exposure to the animals during transportation. Therefore we appreciate the reviewer for critically analyzing this and providing appropriate scientific reason for consistent RT values. The same explanation has been included in the discussion section of the revised manuscript to maintain consistent explanation for the results obtained on RT, RR and PR.

Comment 9: And the authors concluded that: “The findings highlight the practicality of prebiotic and herbal supplements as non-invasive, cost-effective strategies to improve animal welfare and productivity during transportation, particularly in challenging environmental conditions. These results provide a scientific basis for incorporating such supplements into livestock management practices, especially in arid and semi-arid regions where stressors are exacerbated by high temperatures.”(Lines 413 to 418). After all, were the animals subjected to heat stress or not? The answer must be well-founded in the characterization of the THI in which the animals were transported.

Response: We appreciate the reviewer for this critical comment. With the inclusion of THI table, result and discussion these statements in the conclusion was justified. The changes are reflected in lines 198-203; 267-275 and table 3 of the revised manuscript.

Comment 10: Therefore, although the researchers obtained good results with the research, the manuscript needs to be improved in the aspects mentioned.

Response: We appreciate the reviewers for his critical comments to improve the manuscript. We have done our best to address all the comments of this reviewer appropriately.

Reviewer 4 Report

Comments and Suggestions for Authors

Reviewer comments for manuscript ID vetsci-3572895 entitled ‘The Role of Prebiotic and Herbal Supplementation in Enhancing Welfare and Resilience of Kenguri Sheep subjected to Transportation Stress’

General Comments

Transportation stress in animals is a concern for animal welfare. Biomarkers for stress have been analysed through different biological fluids like blood,serum/plasma, saliva and urine and even in faeces and hair. Supplementation of anti-stressor compounds like prebiotics, probiotics and herbal preparations , in research studies and further in routine veterinary therapeutics have been used contemporarily. The efficacy of these supplements still needs to be tested through research and clinical studies as these supplements are claimed to be having long term advantages with no residues in animal body.

The present study on the role of prebiotics and herbal supplementation in Kenguri sheep to ameliorate the transportation stress is a valuable addition to the growing body of literature on transport stress amelioration. The manuscript is nicely written but almost no errors. Introduction section is very nicely presented with identification of research gaps and formulation of hypothesis. Results are presented concisely, well supported with tables for better understanding of the authors.

I have some clarification and doubts on the methodology that I specifically pointed out. One time supplementation of prebiotics and herbal supplements almost an hour before transportation is doubtful as even commercially available anti-stress supplements are advised to be administered for at least a week to obtain tangible results. The sheep transported in this study were not subject to heat stress as evident from the data and findings that render the interpretation of results questionable. A number of times in the manuscript, non-significant findings ( numerical changes in the data during experimentation) have been used to analyse the result. This will lead to error prone interpretation of results.

I am sorry, I need clarifications/ corrections  on these points before I recommend the publication of the manuscript.

Specific comments

Lines 44-45: Please elaborate the uniqueness.

Lines 112-13: Why this particular age group and body weight sheep were selected for the study? Livestock transport is usually of adult animals weighing 30-40 kg. As the study envisages simulation of live transport effects, I feel the age and weight of the sheep should have taken into consideration. Please clarify.

Line 118: Is one time supplementation enough to ameliorate the transportation stress or a longer period (a week prior to the date of transportation) is needed?  What was the physiology behind feeding 45-60 minutes before transportation? Please clarify.

Lines 280-91: The ambient temperature range during the transportation was below the heat stress challenge for the sheep. To simulate the heat stress due to transportation, don’t you feel the experiment should have been conducted at a higher ambient temperature range? Please clarify.

Lines 327-34: I am sorry, non-significant numerical changes in data should not be interpreted to causal  and trend relationships, as these are vulnerable to errors and misinterpretation.

Author Response

Reviewer 4

Comment 1: General Comments. Transportation stress in animals is a concern for animal welfare. Biomarkers for stress have been analysed through different biological fluids like blood,serum/plasma, saliva and urine and even in faeces and hair. Supplementation of anti-stressor compounds like prebiotics, probiotics and herbal preparations , in research studies and further in routine veterinary therapeutics have been used contemporarily. The efficacy of these supplements still needs to be tested through research and clinical studies as these supplements are claimed to be having long term advantages with no residues in animal body.

Response: We appreciate the reviewer for these valuable suggestion. This information also was used in the introduction section in lines 92-97 of the revised manuscript to justify the study.

Comment 2: The present study on the role of prebiotics and herbal supplementation in Kenguri sheep to ameliorate the transportation stress is a valuable addition to the growing body of literature on transport stress amelioration. The manuscript is nicely written but almost no errors. Introduction section is very nicely presented with identification of research gaps and formulation of hypothesis. Results are presented concisely, well supported with tables for better understanding of the authors.

Response: We thank the reviewer for his appreciation of our work as well as for his encouraging comments.

Comment 3: I have some clarification and doubts on the methodology that I specifically pointed out. One time supplementation of prebiotics and herbal supplements almost an hour before transportation is doubtful as even commercially available anti-stress supplements are advised to be administered for at least a week to obtain tangible results.

Response: Physiological stress responses, such as changes in cortisol levels, oxidative stress markers, and inflammatory responses, can occur rapidly after exposure to a stressor or an intervention. Several studies have demonstrated that oral supplementation with nutritional supplements (bioactives or adaptogens) can lead to measurable changes in antioxidant enzyme activities and stress markers within 4–8 hours of administration. Therefore, it is plausible that a single dose can exert detectable physiological effects within an 8-hour window, especially in the context of acute stress such as transportation. This information was included in the revised manuscript in lines 284-290.

Comment 4: The sheep transported in this study were not subject to heat stress as evident from the data and findings that render the interpretation of results questionable.

Response: We would like to clarify that although we measured weather variables to calculate temperature humidity index (indicated in materials & methods) we did not include this in results in original manuscript. Now in the results and discussion the THI results were included and discussed indicating extreme heat stress for animals during transportation. These information are can be found in lines 198-203; 267-275 and table 3 of the revised manuscript. Therefore, we believe that our interpretation of results was appropriate.

Comment 5: A number of times in the manuscript, non-significant findings (numerical changes in the data during experimentation) have been used to analyse the result. This will lead to error prone interpretation of results.

Response: We agree it was a mistake to explain non-significant findings. We rectified this mistake throughout the results and discussion section of the revised manuscript.

Comment 6: I am sorry, I need clarifications/ corrections on these points before I recommend the publication of the manuscript.

Response: We have addressed all the above comments raised by this reviewer. We believe the responses to those comments were appropriate.

Comment 7: Specific comments. Lines 44-45: Please elaborate the uniqueness.

Response: This sentence was elaborated to explain the uniqueness of small ruminants getting more affected by transportation stress. The information can be found in lines 46-52 of the revised manuscript.

Comment 8: Lines 112-13: Why this particular age group and body weight sheep were selected for the study? Livestock transport is usually of adult animals weighing 30-40 kg. As the study envisages simulation of live transport effects, I feel the age and weight of the sheep should have taken into consideration. Please clarify.

Response: We would like to clarify that these animals are procured for breeding purpose and hence we want to get young flock and rear them for the said purpose in our institute. Hence we went for less than one year old animals. 

Comment 9: Line 118: Is one time supplementation enough to ameliorate the transportation stress or a longer period (a week prior to the date of transportation) is needed?  What was the physiology behind feeding 45-60 minutes before transportation? Please clarify.

Response:   Physiological stress responses, such as changes in cortisol levels, oxidative stress markers, and inflammatory responses, can occur rapidly after exposure to a stressor or an intervention. Several studies have demonstrated that oral supplementation with nutritional supplements (bioactives or adaptogens) can lead to measurable changes in antioxidant enzyme activities and stress markers within 4–8 hours of administration. Therefore, it is plausible that a single dose can exert detectable physiological effects within an 8-hour window, especially in the context of acute stress such as transportation. This information was included in the revised manuscript in lines 284-290.

Comment 10: Lines 280-91: The ambient temperature range during the transportation was below the heat stress challenge for the sheep. To simulate the heat stress due to transportation, don’t you feel the experiment should have been conducted at a higher ambient temperature range? Please clarify.

Response: We agree to this point of reviewer. However we would like to clarify that even such temperature can be stressful inside the transport vehicle in an enclosure.  This can be evident from the THI programed based on the weather variables recorded. The THI values obtained was extremely stressful inside the vehicle and moderately stressful outside the vehicle during the transportation. Now in the results and discussion the THI results were included and discussed indicating extreme heat stress for animals during transportation. These information can be found in lines in lines 198-203; 267-275 and table 3 of the revised manuscript.

Comment 11: Lines 327-34: I am sorry, non-significant numerical changes in data should not be interpreted to causal and trend relationships, as these are vulnerable to errors and misinterpretation.

Response: We completely agree with the reviewer and we accept it is not correct to give importance to non-significant effects. These sentences were removed in the revised manuscript. In other places also the non-significant effect statements were reduced in the revised manuscript.

Reviewer 5 Report

Comments and Suggestions for Authors

Review report of "The Role of Prebiotic and Herbal Supplementation in Enhancing Welfare and Resilience of Kenguri Sheep subjected to Transportation Stress"

Comments to authors:

Overall, the article addresses a highly relevant topic in transport stress mitigation by investigating the efficacy of a prebiotic or herbal supplement and their combination. It demonstrates good scientific merit, with comprehensive methodologies and robust experimental data supporting its claims. Still, some revisions are necessary.

In general, the abstract would benefit from the inclusion of quantitative data, as it currently appears overly descriptive. For example, the sentence in L22, need to add some quantitative values of reduced magnitude of RR and PR.

L26. Abbreviations should be defined at their first occurrence.

Better keywords should be chosen. As a general rule, keywords should differ from the words already present in the title and abstract. This improves searchability and discoverability in databases.

L136. Need to add more specific of weather variable. The instrument used, the frequency of recording, etc.

L148. Need to add a supporting  reference for the rectal temp. measurement.

Table 3. Probably change p value of 0.000 to “< 0.001”

Suggest to add a separate column before p value column and report the SEM values in that column. Follow this for other tables as well.

Table 4. remove “AT” that has been redundantly repeated, and perhaps juts explain in table footnote. Or somewhere in the title.

Finally, authors need to provide a statement if any AI generative tool was used during the preparation of this manuscript.

Author Response

Reviewer 5

Comment 1: Overall, the article addresses a highly relevant topic in transport stress mitigation by investigating the efficacy of a prebiotic or herbal supplement and their combination. It demonstrates good scientific merit, with comprehensive methodologies and robust experimental data supporting its claims. Still, some revisions are necessary.

Response: We appreciate the reviewer for the encouraging comments on our work. We have also addressed all the comments of this reviewer to improve the quality of the paper.

Comment 2: In general, the abstract would benefit from the inclusion of quantitative data, as it currently appears overly descriptive. For example, the sentence in L22, need to add some quantitative values of reduced magnitude of RR and PR.

Response: This point raised by the reviewer was well taken and accordingly values were entered for statistically significant results in the revised manuscript in lines 20-28.

Comment 3: L26. Abbreviations should be defined at their first occurrence.

Response: We thank the reviewer for pointing out this mistake and accordingly all the abbreviations were expanded in the abstract in lines 22; 24; 28-29 of the revised manuscript.

Comment 4: Better keywords should be chosen. As a general rule, keywords should differ from the words already present in the title and abstract. This improves searchability and discoverability in databases.

Response: We agree with the reviewer on this particular point and accordingly the keywords were changed in line 34 of the revised manuscript .

Comment 5: L136. Need to add more specific of weather variable. The instrument used, the frequency of recording, etc.

Response:  This suggestion of the reviewer was incorporated in the revised manuscript in lines 150-153.

Comment 6: L148. Need to add a supporting reference for the rectal temp. measurement.

Response: This suggestion of the reviewer was incorporated in the revised manuscript in line 165.

Comment 7: Table 3. Probably change p value of 0.000 to “< 0.001”

Response: This suggestion was incorporated and can be found in table 4 of the revised manuscript.

Comment 8: Suggest to add a separate column before p value column and report the SEM values in that column. Follow this for other tables as well.

Response: This suggestion was incorporated and can be found in table 4, table 5 and table 6 of the revised manuscript.

Comment 9: Table 4. remove “AT” that has been redundantly repeated, and perhaps juts explain in table footnote. Or somewhere in the title.

Response: This suggestion was incorporated and can be found in table 5 of the revised manuscript. The abbreviation was expanded in the footnote of the same table.

Comment 10: Finally, authors need to provide a statement if any AI generative tool was used during the preparation of this manuscript.

Response: We thank the reviewer for this suggestion. We would like to clarify that no AI tools were used to prepare this manuscript. Accordingly, a statement on this aspect was included after conclusion in lines 496-497 of the revised manuscript.

Round 2

Reviewer 2 Report

Comments and Suggestions for Authors

Authors have attended my comments.

Author Response

Comment 1: Authors have attended my comments

Response: We thank the reviewer for acknowledging our revision

Reviewer 3 Report

Comments and Suggestions for Authors

Dear authors,

I present my considerations about the manuscript, making the descriptions as bellow:

Now, the main question is well addressed by the research. It is the effects of the use of prebiotic and herbal supplements on welfare and resilience of female sheeps subjected to thermal heat stress during transportation.

The researchers obtained good results with the research and the manuscript was improved compared to the previous version in all topics (abstract, introduction; Keywords, materials and methods, results, discussion and conclusions).

Still, small adjustments are necessary as bellow:

Line 34: Put the keywords in alphabetical order.

Line 203: goat must be replaced by sheep.

Line 519: Withania Somnifera must be replaced by Withania somnifera and in italics.

Line 522: Curcuma Longa L. must be replaced by Curcuma longa L. Pay attention: letter L. is not in italics.

Line 526: Aloe Vera must be replaced by Aloe vera and in italics.

Line 527: in vitro must be in italics.

Line 550: gynura bicolor must be replaced by Gynura bicolor and in italics.

Line 567: Withania Somnifera must be replaced by Withania somnifera.

 Line 576: Morinda Citrifolia must be replaced by Morinda citrifolia.

Line 602: Withania Somnifera must be replaced by Withania somnifera.

Author Response

Comment 1: It is the effects of the use of prebiotic and herbal supplements on the welfare and resilience of female sheep subjected to thermal heat stress during transportation.

Response: We thank the reviewer for this suggestion. Accordingly, the title has been modified in the revised manuscript.

Comment 2: The researchers obtained good results with the research, and the manuscript was improved compared to the previous version in all topics (abstract, introduction, Keywords, materials and methods, results, discussion, and conclusions).

Response: We thank the reviewer for encouraging comments on the revised manuscript.

Comment 3: Line 34: Put the keywords in alphabetical order.

Response: The keywords were rearranged alphabetically in the revised manuscript.

Comment 4: Line 203: goat must be replaced by sheep.

Response: This mistake was corrected in table 3 of the revised manuscript.

Comment 5: Line 519: Withania Somnifera must be replaced by Withania somnifera and in italics.

Response: This mistake was corrected in reference number 9 of the revised manuscript.

Comment 6: Line 522: Curcuma Longa L. must be replaced by Curcuma longa L. Pay attention: letter L. is not in italics.

Response: This mistake was corrected in reference number 10 of the revised manuscript.

Comment 7: Line 526: Aloe Vera must be replaced by Aloe vera and in italics.

Response: This mistake was corrected in reference number 11 of the revised manuscript.

Comment 8: Line 527: in vitro must be in italics.

Response: This mistake was corrected in reference number 11 of the revised manuscript.

Comment 9: Line 550: gynura bicolor must be replaced by Gynura bicolor and in italics.

Response: This mistake was corrected in reference number 22 of the revised manuscript.

Comment 10: Line 567: Withania Somnifera must be replaced by Withania somnifera.

Response: This mistake was corrected in reference number 29 of the revised manuscript.

Comment 11:  Line 576: Morinda Citrifolia must be replaced by Morinda citrifolia.

Response: This mistake was corrected in reference number 33 of the revised manuscript.

Comment 12: Line 602: Withania Somnifera must be replaced by Withania somnifera.

Response: This mistake was corrected in reference number 44 of the revised manuscript.

Reviewer 5 Report

Comments and Suggestions for Authors

I have re-reviewed the revised manuscript and believe the author has addressed the comments appropriately. I have no further suggestions and recommend the manuscript for publication.

Author Response

Comment 1: I have re-reviewed the revised manuscript and believe the author has addressed the comments appropriately. I have no further suggestions and recommend the manuscript for publication

Response: We thank the reviewer for acknowledging our revision